# THE CLOSER, THE BETTER: TOWARDS BETTER REPRESENTATION LEARNING FOR FEW-SHOT CLASS-INCREMENTAL LEARNING

## ABSTRACT

Aiming to incrementally learn new classes with only few samples while preserving the knowledge of base (old) classes, few-shot class-incremental learning (FSCIL) faces several challenges, such as overfitting and catastrophic forgetting. To bypass the issues, many works have employed a non-parametric classifier: representing each class with the average of features obtained with a fixed feature extractor trained on base classes. Under such formulation, representation learning is critical to tackle the unique challenges of FSCIL: (1) the transferability of the learned representation to new knowledge, (2) the discriminability between all classes, regardless of old or new. Recent advances in representation learning, such as contrastive learning, have greatly improved the transferability, which is often attributed to the spread of intra-class features. However, we observe that solely improving the transferability can harm the discriminability of FSCIL models, as too much spread of features can degrade the quality of the feature-mean class representation. Upon the observation and further experimental analysis, we claim that not only we need to increase the intra-class distance, but we also need to *decrease* the inter-class distance. Trying to secure the spread of features and discriminability within a more confined space due to small inter-class distances, the learned representation strikes a good balance between the transferability and discriminability. The strong performance, without any weight update while learning new classes, demonstrates the effective discriminability and transferability of our new representation, founded upon our seemingly counter-intuitive claim: **the-Closer-the-Better (CnB)**.

## 1 INTRODUCTION

Owing to its strong representation power, deep neural networks (DNNs) boast outstanding performance across various fields. However, such feat comes at the cost of a tremendous amount of human efforts and time to collect an immense amount of data with accurate annotation. The data hunger of DNNs poses a challenge especially under dynamic real-world environments, where DNNs are required to learn new concepts with few examples, while retaining previously learned concepts. Humans, on the other hand, are capable of obtaining new knowledge with only a small number of examples without forgetting prior knowledge. In an effort to bridge the gap, few-shot class-incremental learning (FSCIL) (Tao et al., 2020) aims to design artificial intelligence systems that can learn new classes with few examples while maintaining the performance on previously seen classes.

To achieve the goal of FSCIL, we need to address the catastrophic forgetting (forgetting of previous knowledge while learning new concepts) (Delange et al., 2021; Kirkpatrick et al., 2017) and overfitting issues (overfitting to few examples and thus poor generalization) (Koch et al., 2015; Vinyals et al., 2016). To bypass this convoluted mixture of issues, which is the unique challenge of FSCIL, most previous works (Zhang et al., 2021; Hersche et al., 2022; Yang et al., 2023; Zou et al., 2022; Zhou et al., 2022a; Peng et al., 2022a) fix the parameters of a feature extractor after training it on base (old) classes. With the fixed feature representation, the recent FSCIL methods employ non-parametric classifier, using the class prototype representation (Snell et al., 2017) for both base and new classes by taking the average of features from examples belonging to each class. Such formulation strongly relies on the representation learned from base classes. Thus, while freezing the feature extractor may prevent catastrophic forgetting and overfitting, the models are not able to learn new concepts, due

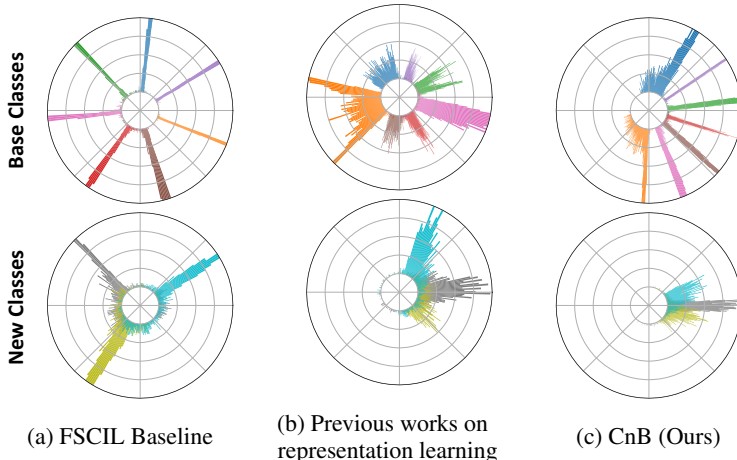

(a) FSCIL Baseline  (b) Previous works on representation learning  (c) CnB (Ours)

Figure 1: Visualization of representation trained on MNIST[1]. **(a) FSCIL baseline** (Zhang et al., 2021; Hersche et al., 2022): exhibits great base-class discriminability (large inter-class distance) but weak transferability on new classes (collapse to base classes). **(b) Previous works on representation learning** (Kornblith et al., 2021; Islam et al., 2021; Chen et al., 2022): benefit new classes (less collapse to base classes), while compromising base-class discriminability in the context of FSCIL problem (too much spread). **(c) CnB (Ours)**: Along with spreading the representation, reducing inter-class distance achieves base class discriminability (not too much spread) and further enhances transferability (even less overlap between base classes and new classes).

to the lack of transferability of the learned representation to new classes. While many works have focused on updating classifier weights (Zhang et al., 2021; Kalla & Biswas, 2022; Yang et al., 2023) to address the problem, we shift our attention to the representation itself.

There has been great advances, particularly contrastive learning (Chen et al., 2020; He et al., 2020), in representation learning to improve the transferability of learned representation to downstream tasks. A few works (Islam et al., 2021; Chen et al., 2022) have attributed the strong transferability of contrastive learning to the "spread out" of features. The spread of features leads to the emphasis of low- and mid-level features, which are more transferable to and can be better shared by new tasks, in contrast to less transferable high-level features learned by the optimization of softmax cross-entropy (SCE) loss (Islam et al., 2021). Islam et al. (2021) further show that the joint combination of SCE loss and self-supervised contrastive (SSC) loss can lead to better transferability of the learned representation. Kornblith et al. (2021) have also found the relationship between the temperature of SCE loss and the extent of intra-class variation (a.k.a. the spread of features), suggesting that lower temperature leads to better transferability. The previous findings on representation learning suggest that the joint optimization of SCE loss with lower temperature and SSC may benefit FSCIL.

Despite the foregoing, we find that the joint objective of SCE loss with lower temperature and SSC is not enough to find good representation for FSCIL; in fact, it harms the performance on base classes. Based on our experimental analysis, we observe that the joint objective leads to too much feature spread, which is very detrimental to recent FSCIL methods, since they employ the feature-average class prototype representation. Thus, for FSCIL, where we need to maintain the performance on old classes while incrementally learning new classes, we cannot just focus on the transferability; we also need to focus on maintaining the discriminability, especially with the feature-average class representation that does not favor the spread of features.

In this work, with the support from our observation in Figure 1, we argue that a large spread of features is caused by a large inter-class distance, which is encouraged by SCE loss and preferred by previous works. That is, features are more spread out with larger inter-class distance, since features are encouraged to be shared among classes for transferability (Islam et al., 2021), Thus, in contrast to common beliefs and practices, we claim the large inter-class distance is detrimental to FSCIL and thereby propose to *decrease* the inter-class distance. Reducing the inter-class distance leads

---

[1]Following Liu et al. (2020); Zheng et al. (2018); Wang et al. (2018), we train a network with 2-D feature dimension and visualize using angular histogram. Each color indicates a different class.

to more compact feature space, which facilitates both the sharing of features and the learning of fine-grained differences between classes now that different class features are placed closer. As such, we propose a new objective for FSCIL towards representation learning with better transferability *and* discriminability: the joint objective of SCE loss with lower temperature, SSC loss, and the inter-class distance minimization. Our experimental results demonstrate that our newly learned representation, named **the-Closer-the-Better (CnB)**, results in better transferability and discriminability without storing exemplars, test-time augmentation, nor any weight update when learning new classes.

## 2 PROPOSED METHOD

### 2.1 BACKGROUND: PROBLEM FORMULATION

Following the formulation of few-shot class incremental learning (FSCIL) from Tao et al. (2020), we assume a sequence of training sessions with its corresponding datasets $\{\mathcal{D}^{(0)}, \mathcal{D}^{(1)}, \cdots, \mathcal{D}^{(T)}\}$. $\mathcal{D}^{(t)} = \{(\boldsymbol{x}_i^{(t)}, y_i^{(t)})\}_{i=1}^{N_t}$ consists of $N_t = |\mathcal{D}^{(t)}|$ number of training examples $\boldsymbol{x}_i^{(t)}$ with its class label $y_i^{(t)} \in \mathcal{C}^{(t)}$ (for the sake of simplicity, we will exclude the superscript of both $\boldsymbol{x}_i^{(t)}$ and $y_i^{(t)}$), where $\mathcal{C}^{(t)}$ is the set of classes in its respective dataset $\mathcal{D}^{(t)}$ and $\mathcal{C}^{(s)} \cap \mathcal{C}^{(t)} = \emptyset$ for $s \neq t$ (each dataset has its own distinct classes with no overlap). In the first session (a.k.a. base session) with the dataset $\mathcal{D}^{(0)}$, there is assumed to be a large number of classes available with an abundant amount of training data for each class. In the subsequent sessions (a.k.a. incremental sessions) with the datasets $\mathcal{D}^{(\geq 1)}$, each dataset is assumed to have few training examples for each class. In particular, FSCIL is said to have a $N$-way $K$-shot setting when each incremental session has $N$ classes with $K$ examples for each class. At each $t$-th training session, only its corresponding dataset $\mathcal{D}^{(t)}$ is accessible for training. After each $t$-th session, the evaluation is performed on all previously seen classes $\mathcal{C}^{(\leq t)}$ using test datasets $\mathcal{D}_{test}^{(\leq t)}$, which consists of test examples with the class label set $\mathcal{C}^{(\leq t)}$.

### 2.2 BACKGROUND: BASELINE

Typically, a classification network consists of feature extractor $f_{\boldsymbol{\theta}}(\cdot)$ and a classification layer with its weights $\phi$, where $d$ is the feature dimension. The training objective in the base session is simply the softmax cross-entropy loss with the cosine similarity $\text{sim}(\cdot, \cdot)$ as logits (Deng et al., 2019):

$$\mathcal{L}_{ce} = \frac{1}{B} \sum_{i=1}^{B} -\log \frac{\exp(\frac{1}{\tau} \text{sim}(\boldsymbol{z}_i, \boldsymbol{\phi}_{y_i}))}{\sum_{j=1}^{|\mathcal{C}^{(0)}|} \exp(\frac{1}{\tau} \text{sim}(\boldsymbol{z}_i, \boldsymbol{\phi}_{y_j}))}, \tag{1}$$

where $\boldsymbol{z}_i = f_{\boldsymbol{\theta}}(\boldsymbol{x}_i)$; $B$ is the batch size and $\tau$ is the temperature parameter. Incrementally updating weights with few examples in incremental sessions can make the network vulnerable to both catastrophic forgetting and overfitting. In order to bypass the problems, several works (Zhang et al., 2021; Hersche et al., 2022) suggest minimizing weight updates by freezing the feature extractor after the base session and using feature-average class prototype representation (Snell et al., 2017). Specifically, after the base session, trained $\phi$ is replaced with class prototypes, a process we refer to as classifier replacement (CR), and new class prototypes are obtained in the subsequent incremental sessions. The $i$-th class prototype is acquired by averaging the features of training samples of the $i$-th class:

$$\boldsymbol{\phi}_{c_i}^P = \frac{1}{N_{c_i}} \sum_{(\boldsymbol{x}_j, y_j) \in \mathcal{D}^{(\geq 0)}} \mathbb{1}_{[y_j = c_i]} f_{\boldsymbol{\theta}}(\boldsymbol{x}_j), \tag{2}$$

where $N_{c_i}$ is the number of training samples associated with the $i$-th class and $\mathbb{1}_{[\cdot]}$ is an indicator function whose value is 1 if the subscript condition is $\text{True}$ and 0 otherwise. For an input $\boldsymbol{x}$, the classification score for the $i$-th class is computed by $\text{sim}(f_{\boldsymbol{\theta}}(\boldsymbol{x}), \boldsymbol{\phi}_{c_i}^P)$.

While this simple baseline bypasses the forgetting and overfitting issues, it heavily relies on the quality of the representation trained solely on the base classes. Consequently, the primary focus of this paper is to investigate the important factors influencing representation learning for FSCIL and strategies for improving them.

### 2.3 TRANSFERABILITY, FEATURE SPREAD, AND ITS ADVERSE EFFECTS ON FSCIL

As shown in Figure 1a, our FSCIL baseline exhibits a narrow feature intra-class distribution, which is widely perceived as representation collapse (Papyan et al., 2020). Recent studies (Chen et al.,

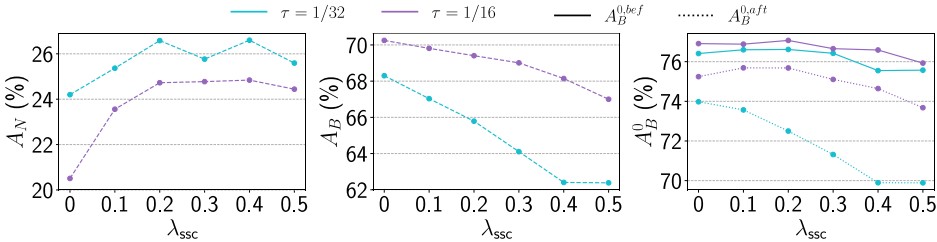

Figure 2: The impact of the spread of representation. Stronger emphasis on self-supervised contrastive loss (larger $\lambda_{\text{ssc}}$) and low temperature (skyblue) enhances the new-class performance $A_N$ (**left**), but at the expense of base-class performance $A_B$ (**center**) both before and after the classifier replacement ($A_B^{0,\text{bef}}$ and $A_B^{0,\text{aft}}$, respectively) (**right**).

2022; Islam et al., 2021; Xue et al., 2023) have revealed that the collapsed representation shows poor transferability due to the loss of shareable low- and mid-level features that new classes can benefit from. As such, they suggest the joint optimization with a self-supervised contrastive task (Chen et al., 2020; He et al., 2020), which promotes the spread of features and thus sharing of features among classes. The self-supervised contrastive learning optimizes the infoNCE loss (van den Oord et al., 2018), regarding an augmented view from a query image as positive and the other images as negative samples. The self-supervised contrastive (SSC) loss for a positive pair $(i, j)$ is:

$$\mathcal{L}_{\text{ssc}}^{(i,j)} = -\log \frac{\exp(\frac{1}{\tau}\text{sim}(\boldsymbol{z}_i, \boldsymbol{z}_j))}{\sum_{k=1}^{B} \mathbb{1}_{[k \neq i]} \exp(\frac{1}{\tau}\text{sim}(\boldsymbol{z}_i, \boldsymbol{z}_k))}, \tag{3}$$

where $B$ is the total number of samples, including augmented images, and $\boldsymbol{z}_j$ is the feature from an augmented view of $\boldsymbol{x}_i$. The loss is averaged for every positive pair in a batch.

In parallel, several studies have proposed reducing the temperature and margin parameters in the softmax cross-entropy loss as a another way to encourage feature sharing (Liu et al., 2020; Zou et al., 2022; Kornblith et al., 2021). Our empirical analysis, detailed in Appendix C, demonstrates that the temperature parameter exerts a more substantial influence on the transferability when compared to the margin. Thus, we focus on lowering temperature $\tau$ to further enhance the transferability. Indeed, we observe that the combination of $\mathcal{L}_{\text{ssc}}$ and low $\tau$ results in better transferability for FSCIL, as demonstrated in Figure 2, and larger spread of features displayed in Figure 1b.

Nevertheless, as illustrated in the center figure in Figure 2, we observe that the pursuit of better transferability still results in an unavoidable degradation in discriminability on the base classes. While exploring the dilemma, we discover that the performance decline on the base classes mainly arises from the base class classifier replacement (CR) strategy, which is elaborated in Section 2.2. The right figure in Figure 2 shows the accuracy of the base classes subsequent to the base session training ($A_B^0$), without cosidering the new classes, both before and after CR, denoted by $A_B^{0,\text{bef}}$ and $A_B^{0,\text{aft}}$, respectively. In the absence of CR, $A_B^{0,\text{bef}}$ exhibits a relatively high value and a small variance as $\tau$ and $\lambda_{\text{ssc}}$ vary (indicated by the solid lines). Conversely, $A_B^{0,\text{aft}}$ is considerably lower than $A_B^{0,\text{bef}}$ with relatively higher variance (indicated by the dotted lines). Considering CR is suggested for improving transferability of the classifier by replacing the trained weights specifially tailored to the base classes, these findings suggest that the trade-off between transferability and discriminability is aggravated in the context of the FSCIL problem. Therefore, it is essential to delve into representation learning tailored specifically for the FSCIL problem.

## 2.4 REDUCING INTER-CLASS DISTANCE MATTERS

As discussed in Section 2.3, learning shareable features through representation spreading proves advantageous for transferability. Nevertheless, we find the encouraging representation spread greatly exacerbates the trade-off between transferability and discriminability. Based on the observation in Figure 1b, we hypothesize that such dilemma appears to arise from large inter-class distance since the representation spreading factors could push features into the extensive inter-class space, which may dilute the information on base classes. Furthermore, we argue that the large inter-class distance may impede the effective feature sharing among classes, thereby undermining transferability. Consequently, in order to retain the knowledge on base classes while promoting effective feature

sharing, we introduce a novel loss function that minimizes the inter-class distance:

$$\mathcal{L}_{\text{inter}} = -\frac{1}{\sum_{i=1}^{B}\sum_{j>i}^{B}\mathbb{1}_{[y_i \neq y_j]}}\sum_{i=1}^{B}\sum_{j>i}^{B}\mathbb{1}_{[y_i \neq y_j]}\texttt{sim}(\boldsymbol{z}_i, \boldsymbol{z}_j). \tag{4}$$

Figure 1c displays that the extent of spread of intra-class features is well regulated by applying $\mathcal{L}_{\text{inter}}$ and Figure 3 demonstrates the performance decline originating from CR is greatly alleviated by minimizing $\mathcal{L}_{\text{inter}}$ with larger loss weights (indicated by the purple line). Moreover, the results indicated by the skyblue line show that reducing inter-class distance significantly improves the performance on the new classes, corroborating our hypothesis.

Our assertion initially seems counter-intuitive, diverging from the common belief of prior works (Yang et al., 2023; Hersche et al., 2022; Zhou et al., 2022a; Song et al., 2023) that maximizing inter-class distance may be beneficial in reserving representation space for future new classes. To further validate the efficacy of reducing inter-class distance on transferability, we propose a measure to quantify how new class samples are distant from base class clusters. We define the measure as the averaged relative angular distance between a new class sample and its nearest base class prototype with respect to the averaged angular distance among all base class prototype pairs:

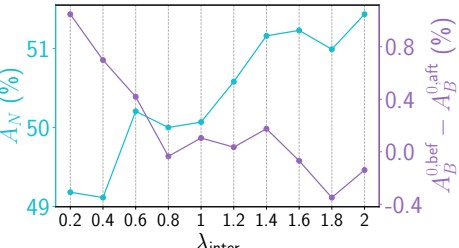

Figure 3: Effect of minimizing inter-class distance. As the weight of $\mathcal{L}_{\text{inter}}$, denoted by $\lambda_{\text{inter}}$, increases, the new-class performance increases (skyblue) and the performance loss from CR is greatly alleviated (purple).

$$\mathcal{T}(f_{\boldsymbol{\theta}}) = \frac{\frac{1}{|\mathcal{D}_{test}^{(>0)}|}\sum_{(\boldsymbol{x}_j,y_j)\in\mathcal{D}_{test}^{(>0)}}\min_{\phi_{base,i}^{P}}\angle(\boldsymbol{z}_j, \phi_{base,i}^{P})}{\sum_{j=1}^{|\mathcal{C}^{(0)}|}\sum_{k>j}^{|\mathcal{C}^{(0)}|}\angle(\phi_j^P, \phi_k^P)\,/\,\binom{|\mathcal{C}^{(0)}|}{2}}, \tag{5}$$

where $\phi_{base,i}^{P}$ and $\angle(\cdot, \cdot)$ indicate the $i$-th base class prototype and the angular distance between two input vectors, respectively. To consider the varying size of representation space depending on methods, the denominator serves as a normalization factor within the angular representation space. If the learned representation of $f_{\boldsymbol{\theta}}$ has a distinguishable representation for the new classes, $\mathcal{T}(f_{\boldsymbol{\theta}})$ could be large. For sanity test, we check the relationship between $\mathcal{T}(f_{\boldsymbol{\theta}})$ and the new class accuracy, which is demonstrated in Figure 4a.

Using this measure, we analyze the relationship between inter-class distance, spread of features, and the transferability of learned representation. To do so, we train the feature extractor using $\mathcal{L}_{\text{ce}}$ with low temperature, $\mathcal{L}_{\text{ssc}}$, and $\mathcal{L}_{\text{inter}}$ with varying loss weights for $\mathcal{L}_{\text{ssc}}$ and $\mathcal{L}_{\text{inter}}$, denoted by $\lambda_{\text{ssc}}$ and $\lambda_{\text{inter}}$, respectively. Figure 4b and Figure 4c show the relationship between the inter-class distance, $\mathcal{T}(f_{\boldsymbol{\theta}})$, and the performance on the new classes. The results demonstrate that the joint optimization of $\mathcal{L}_{\text{ssc}}$ and $\mathcal{L}_{\text{inter}}$ leads to increase in both $\mathcal{T}(f_{\boldsymbol{\theta}})$ and $A_N$, indicating that smaller inter-class distances actually lead to more discriminability between base classes and new classes. The analysis corroborates our seemingly counter-intuitive hypothesis that the closer classes are, the better.

In summary, we have observed that the spread of representation achieved by lowering temperature and self-supervised contrastive learning is advantageous for learning transferable representation. However, spread representation itself cannot address the trade-off between transferability for new classes and discrimiability for base classes in the context of FSCIL. Our analysis demonstrates that decreasing inter-class distance enhances discriminability by regularizing the intra-class spread and improves the transferability by promoting the effective learning of shareable information among classes when combined with the feature-spread-encouraging loss. Consequently, our final loss function is the combination of cross-entropy loss with lower temperature, self-supervised contrastive loss, and inter-class distance minimizing loss:

$$\mathcal{L} = \mathcal{L}_{\text{ce}} + \lambda_{\text{ssc}}\mathcal{L}_{\text{ssc}} + \lambda_{\text{inter}}\mathcal{L}_{\text{inter}}. \tag{6}$$

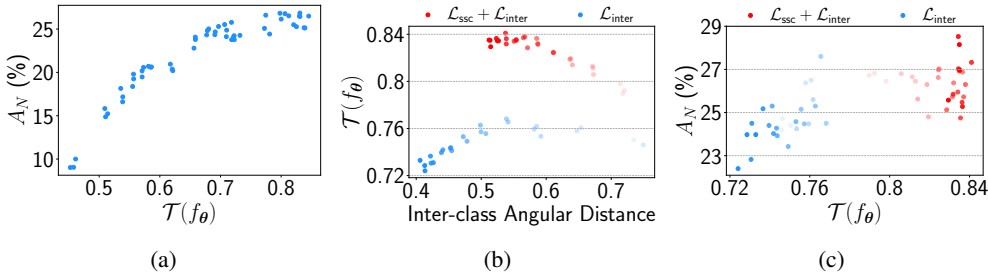

(a)  (b)  (c)

Figure 4: **(a) Sanity test for $\mathcal{T}(f_{\boldsymbol{\theta}})$**: $\mathcal{T}(f_{\boldsymbol{\theta}})$ has a positive correlation with the performance on the new classes. Each data point is obtained by different configurations of $\tau$ and $\lambda_{\mathrm{ssc}}$ (without $\mathcal{L}_{\mathrm{inter}}$). **(b),(c) Relationship between inter-class distance, $\mathcal{T}(f_{\boldsymbol{\theta}})$, and $A_N$**: Integrated with the representation spreading, reducing inter-class distance encourages better transferability (red points). However, the tendency is broken when reducing inter-class distance without representation spreading (blue points). The more transparent dots indicate smaller $\lambda_{\mathrm{inter}}$. We fix $\lambda_{\mathrm{ssc}}$ as 0.1 if it is used.

## 3 EXPERIMENTS

### 3.1 EXPERIMENTAL DETAILS

**Dataset.** Following the benchmark settings proposed in Tao et al. (2020), we evaluate the proposed method on CIFAR100 (Krizhevsky & Hinton, 2009), *mini*ImageNet (Vinyals et al., 2016), and CUB200 (Wah et al., 2011). For CIFAR100 and *mini*Imagenet, the total number of classes is 100: 60 base classes and 40 new classes. The 40 new classes are split into 8 disjoint sets of 5 classes, each set of which is sequentially provided with 5 training examples per class (5-way 5-shot) in each incremental session. As for CUB200, there total number of classes is 200, with 100 base classes and 100 new classes. 100 new classes are split into 10 disjoint sets of 10 classes, each set of which are sequentially provided with 5 training examples per class (10-way 5-shot) in each incremental session.

**Implementation.** Following the setting from Zhang et al. (2021), we use ResNet-20 (He et al., 2016) for CIFAR100 experiments and ResNet-18 (He et al., 2016) for both *mini*ImageNet and CUB200 experiments. We follow the conventions to use the ResNet-18 model pre-trained on the ImageNet dataset (Russakovsky et al., 2015) for CUB200 experiments. The temperature parameter $\tau$ for the baseline method is $1/16$ and 'low temperature' in the following sections indicates $\tau = 1/32$. We set $\lambda_{\mathrm{ssc}}$ as 0.1, 0.1, and 0.01 for CIFAR100, *mini*ImageNet, and CUB200, respectively, and $\lambda_{\mathrm{inter}}$ as 1, 0.5, and 1.5 for CIFAR100, *mini*ImageNet, and CUB200, respectively. These hyper-parameters are searched via validation using synthesized validation sets. More details including optimization, self-supervised contrastive learning, and the hyper-parameter search strategy are elaborated in Appendix A.

**Evaluation.** We use the accuracy on base ($A_B$), new ($A_N$), and the whole classes ($A_W$) as metrics to evaluate the discriminability and transferability of learned representation. Additionally, we use the performance drop (PD) between the accuracy at the end of the base session (session 0) and the accuracy at the last incremental session to evaluate the degree of forgetting old knowledge (stability) and learning new knowledge (plasticity) simultaneously.

### 3.2 COMPARISON WITH THE EXISTING WORKS

We present the performance of the proposed method, dubbed **the-Closer-the-Better (CnB)**, and prior arts on CUB200 (Table 1), CIFAR100 (Table 3), and *mini*ImageNet (Table 4). In particular, CnB achieves state-of-the-art performance on both CUB200 and CIFAR100 datasets, surpassing the results of previous methods by a large margin with respect to $A_W$ and PD. With *mini*ImageNet, the proposed method exhibits substantially higher $A_W$ than the method with the lowest PD and achieves lower PD than the method with the highest $A_W$, which means the proposed method achieves a better balance between the performance on base and new classes. Notably, our learned representation provides such outstanding performance without any assistance of the storage of previous samples (F2M, ERDIL, IDLVQ-C, and CABD), additional computational modules (CEC, CLOM, NC-FSCIL, SAVC, and MetaFSCIL), and test-time data augmentation (S3C and SAVC), suggesting the critical importance of learning effective representations in FSCIL.

Table 1: 10-way 5-shot incremental learning results on CUB200.

| Method | Acc. in each session (%) | | | | | | | | | | | PD ↓ |
|---|---|---|---|---|---|---|---|---|---|---|---|---|
| | 0 | 1 | 2 | 3 | 4 | 5 | 6 | 7 | 8 | 9 | 10 | |
| Baseline | 79.92 | 76.23 | 73.18 | 69.45 | 67.83 | 65.74 | 64.54 | 63.33 | 61.56 | 61.27 | 60.10 | 19.83 |
| TOPIC (Tao et al., 2020) | 68.68 | 62.49 | 54.81 | 49.99 | 45.25 | 41.40 | 38.35 | 35.36 | 32.22 | 28.31 | 26.26 | 42.42 |
| F2M (Shi et al., 2021) | 81.07 | 78.16 | 75.57 | 72.89 | 70.86 | 68.17 | 67.01 | 65.26 | 63.36 | 61.76 | 60.26 | 20.81 |
| CEC (Zhang et al., 2021) | 75.85 | 71.94 | 68.50 | 63.50 | 62.43 | 58.27 | 57.73 | 55.81 | 54.83 | 53.52 | 52.28 | 23.57 |
| IDLVQ-C (Chen & Lee, 2021) | 77.37 | 74.72 | 70.28 | 67.13 | 65.34 | 63.52 | 62.10 | 61.54 | 59.04 | 58.68 | 57.81 | 19.56 |
| ALICE (Peng et al., 2022b) | 77.40 | 72.70 | 70.60 | 67.20 | 65.90 | 63.40 | 62.90 | 61.90 | 60.50 | 60.60 | 60.10 | 17.30 |
| CLOM (Zou et al., 2022) | 79.57 | 76.07 | 72.94 | 69.82 | 67.80 | 65.56 | 63.94 | 62.59 | 60.62 | 60.34 | 59.58 | 19.99 |
| Entropy Reg. (Liu et al., 2022) | 75.90 | 72.14 | 68.64 | 63.76 | 62.58 | 59.11 | 57.82 | 55.89 | 54.92 | 53.58 | 52.39 | 23.51 |
| LIMIT (Zhou et al., 2022b) | 75.89 | 73.55 | 71.99 | 68.14 | 67.42 | 63.61 | 62.40 | 61.35 | 59.91 | 58.66 | 57.41 | 18.48 |
| SAVC (Song et al., 2023) | 81.85 | 77.92 | 74.95 | 70.21 | 69.96 | 67.02 | 66.16 | 65.30 | 63.84 | 63.15 | 62.50 | 19.35 |
| MetaFSCIL (Chi et al., 2022) | 75.90 | 72.41 | 68.78 | 64.78 | 62.96 | 59.99 | 58.30 | 56.85 | 54.78 | 53.82 | 52.64 | 23.26 |
| FACT (Zhou et al., 2022a) | 75.90 | 73.23 | 70.84 | 66.13 | 65.56 | 62.15 | 61.74 | 59.83 | 58.41 | 57.89 | 56.94 | 18.96 |
| NC-FSCIL (Yang et al., 2023) | 80.45 | 75.98 | 72.30 | 70.28 | 68.17 | 65.16 | 64.43 | 63.25 | 60.66 | 60.01 | 59.44 | 21.01 |
| S3C (Kalla & Biswas, 2022) | 80.62 | 77.55 | 73.19 | 68.54 | 68.05 | 64.33 | 63.58 | 62.07 | 60.61 | 59.79 | 58.95 | 20.83 |
| GKEAL (Zhuang et al., 2023) | 78.88 | 75.62 | 72.32 | 68.62 | 67.23 | 64.26 | 62.98 | 61.89 | 60.20 | 59.21 | 58.67 | 20.21 |
| CABD (Zhao et al., 2023) | 79.12 | 75.37 | 72.80 | 69.05 | 67.53 | 65.12 | 64.00 | 63.51 | 61.87 | 61.47 | 60.93 | 18.19 |
| CnB | 79.40 | 75.92 | 73.50 | 70.47 | 69.24 | 67.22 | 66.73 | 65.69 | 64.00 | 64.02 | **63.58** | **15.82** |

## 3.3 ABLATION STUDIES

To verify the efficacy of the individual components in the proposed method, we conduct ablation studies, which are shown in Table 2. The increase in $A_N$ when employing either a lower temperature in the softmax cross-entropy loss (low $\tau$) or a self-supervised contrastive loss ($\mathcal{L}_{ssc}$) affirms the advantage of feature spread in enhancing the transferability of representations. While the transferability can be greatly improved by utilizing both low $\tau$ and $\mathcal{L}_{ssc}$, a substantial decline in base-class performance $A_B$ is observed, as discussed in Section 2.4. The result from the case where all components are utilized demonstrates that this issue can be effectively resolved by minimizing inter-class distance. Furthermore, as discussed in Section 2.4, reducing inter-class distance is observed to improve the transferability, especially when used with the representation spreading methods. The comprehensive results confirm that the ablation studies support our claims and analysis.

## 3.4 VISUALIZATION RESULTS

For qualitative evaluation, we visualize the learned representation trained by different configurations of the proposed losses, which is illustrated in Figure 6. The value of $\mathcal{T}(f_\theta)$ in the right below in each figure is for measuring the transferability of representation. The baseline representation (a) exhibits the base classes characteristics, portrayed by the features of new classes being mapped to those of base classes, also indicated by the low value of $\mathcal{T}(f_\theta)$. The spreading of features (b) largely resolves the overfitting issues, exhibited by larger distances between new classes and base classes; i.e., the increase in $\mathcal{T}(f_\theta)$. Finally, $\mathcal{L}_{inter}$ is employed to compensate the decreased performance on the base

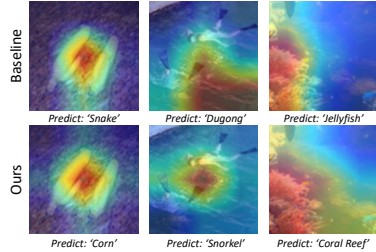

Figure 5: Visualization of class activation map using Chattopadhyay et al. (2018). These examples belong to new classes. Our method correctly predicts the classes, whereas the baseline misclassifies them as base classes.

Table 2: Ablation studies on CIFAR100. 'DA' is the abbreviation for 'data augmentation'.

| DA | low $\tau$ | $\mathcal{L}_{ssc}$ | $\mathcal{L}_{inter}$ | $A_B$ (%) | $A_N$ (%) | $A_W$ (%) | PD |
|---|---|---|---|---|---|---|---|
| ✗ | ✗ | ✗ | ✗ | 68.78 | 14.55 | 47.09 | 25.84 |
| ✓ | ✗ | ✗ | ✗ | 70.13 | 20.95 | 50.46 | 24.81 |
| ✓ | ✗ | ✓ | ✗ | 69.95 | 22.80 | 51.09 | 24.59 |
| ✓ | ✗ | ✗ | ✓ | 69.95 | 20.05 | 49.99 | 26.78 |
| ✓ | ✗ | ✓ | ✓ | 71.43 | 22.83 | 51.99 | 25.26 |
| ✓ | ✓ | ✗ | ✗ | 68.33 | 24.10 | 50.64 | 23.23 |
| ✓ | ✓ | ✓ | ✗ | 66.58 | 25.08 | 49.98 | 23.15 |
| ✓ | ✓ | ✗ | ✓ | 70.17 | 22.40 | 51.06 | 25.27 |
| ✓ | ✓ | ✓ | ✓ | 70.72 | 27.23 | **53.32** | **22.40** |

Table 3: 5-way 5-shot incremental learning results on CIFAR100.

| Method | Acc. in each session (%) | | | | | | | | | PD↓ |
| --- | --- | --- | --- | --- | --- | --- | --- | --- | --- | --- |
| | 0 | 1 | 2 | 3 | 4 | 5 | 6 | 7 | 8 | |
| Baseline | 72.93 | 68.46 | 64.26 | 60.15 | 56.53 | 53.60 | 51.51 | 49.19 | 47.09 | 25.84 |
| ERDIL (Dong et al., 2021) | 73.62 | 68.22 | 65.14 | 61.84 | 58.35 | 55.54 | 52.51 | 50.16 | 48.23 | 25.39 |
| CEC (Zhang et al., 2021) | 73.07 | 68.88 | 65.26 | 61.19 | 58.09 | 55.57 | 53.22 | 51.34 | 49.14 | 23.93 |
| CLOM (Zou et al., 2022) | 74.20 | 69.83 | 66.17 | 62.39 | 59.26 | 56.48 | 54.36 | 52.16 | 50.25 | 23.95 |
| Entropy Reg. (Liu et al., 2022) | 74.4 | 70.2 | 66.54 | 62.51 | 59.71 | 56.58 | 54.52 | 52.39 | 50.14 | 24.26 |
| FACT (Zhou et al., 2022a) | 74.60 | 72.09 | 67.56 | 63.52 | 61.38 | 58.36 | 56.28 | 54.24 | 52.10 | 22.50 |
| LIMIT (Zhou et al., 2022b) | 73.81 | 72.09 | 67.87 | 63.89 | 60.70 | 57.77 | 55.67 | 53.52 | 51.23 | 22.58 |
| MetaFSCIL (Chi et al., 2022) | 74.50 | 70.10 | 66.84 | 62.77 | 59.48 | 56.52 | 54.36 | 49.97 | 48.23 | 24.53 |
| GKEAL (Zhuang et al., 2023) | 74.01 | 70.45 | 67.01 | 63.08 | 60.01 | 57.30 | 55.50 | 53.39 | 51.40 | 22.61 |
| CnB | 75.72 | 71.83 | 68.32 | 64.62 | 61.91 | 59.25 | 57.53 | 55.43 | **53.32** | **22.40** |

Table 4: 5-way 5-shot incremental learning results on *mini*ImageNet.

| Method | Acc. in each session (%) | | | | | | | | | PD↓ |
| --- | --- | --- | --- | --- | --- | --- | --- | --- | --- | --- |
| | 0 | 1 | 2 | 3 | 4 | 5 | 6 | 7 | 8 | |
| Baseline | 72.27 | 67.46 | 63.26 | 59.73 | 56.56 | 53.53 | 50.90 | 48.93 | 47.26 | 25.01 |
| TOPIC (Tao et al., 2020) | 61.31 | 50.09 | 45.17 | 41.16 | 37.48 | 35.52 | 32.19 | 29.46 | 24.42 | 36.89 |
| F2M (Shi et al., 2021) | 67.28 | 63.80 | 60.38 | 57.06 | 54.08 | 51.39 | 48.82 | 46.58 | 44.65 | 22.63 |
| CEC (Zhang et al., 2021) | 72.00 | 66.83 | 62.97 | 59.43 | 56.70 | 53.73 | 51.19 | 49.24 | 47.63 | 24.37 |
| IDLVQ-C (Chen & Lee, 2021) | 64.77 | 59.87 | 55.93 | 52.62 | 49.88 | 47.55 | 44.83 | 43.14 | 41.84 | 22.93 |
| Subspace Reg. (Akyürek et al., 2022) | 80.37 | 73.76 | 68.36 | 64.07 | 60.36 | 56.27 | 53.10 | 50.45 | 47.55 | 32.83 |
| ALICE (Peng et al., 2022b) | 80.6 | 70.6 | 67.40 | 64.50 | 62.50 | 60.00 | 57.80 | 56.80 | **55.70** | 24.90 |
| CLOM (Zou et al., 2022) | 73.08 | 68.09 | 64.16 | 60.41 | 57.41 | 54.29 | 51.54 | 49.37 | 48.00 | 25.08 |
| Entropy Reg. (Liu et al., 2022) | 71.84 | 67.12 | 63.21 | 59.77 | 57.01 | 53.95 | 51.55 | 49.52 | 48.21 | 23.63 |
| LIMIT (Zhou et al., 2022b) | 72.32 | 68.47 | 64.30 | 60.78 | 57.95 | 55.07 | 52.70 | 50.72 | 49.14 | 23.13 |
| MetaFSCIL (Chi et al., 2022) | 72.04 | 67.94 | 63.77 | 60.29 | 57.58 | 55.16 | 52.90 | 50.79 | 49.19 | 22.85 |
| FACT (Zhou et al., 2022a) | 72.56 | 69.63 | 66.38 | 62.77 | 60.60 | 57.33 | 54.34 | 52.16 | 50.49 | 22.07 |
| GKEAL (Zhuang et al., 2023) | 73.59 | 68.90 | 65.33 | 62.29 | 59.39 | 56.70 | 54.20 | 52.59 | 51.31 | 22.28 |
| CABD (Zhao et al., 2023) | 74.65 | 70.43 | 66.29 | 62.77 | 60.75 | 57.24 | 54.79 | 53.65 | 52.22 | 22.43 |
| CnB | 76.02 | 71.61 | 67.99 | 64.69 | 61.70 | 58.94 | 56.23 | 54.52 | 53.33 | 22.69 |

classes due to the spread representation. Reducing inter-class distance also enhances the separability between the new class samples and the clusters of base classes, as evidenced by the further increase in $\mathcal{T}(f_{\boldsymbol{\theta}})$.

We further evaluate our method by visualizing the class activation map using Chattopadhyay et al. (2018), which is illustrated in Figure 5. We observe that the predictions made by the baseline method on new class examples exhibit a significant bias towards the attributes found within base classes, such as color, shape, and background, leading to misclassification into the base classes. For instance, the baseline's prediction of 'Snake' appears to be influenced by the presence of a green, elongated object in the image. Similarly, the predictions of 'Dugong' and 'Jellyfish' seem to be influenced by the ocean background and the resemblance of jellyfish to the coral reef, respectively. On the other hand, the results of our method demonstrate that the learned representation exhibits relatively less bias toward the base classes and enhanced transferability to the new classes.

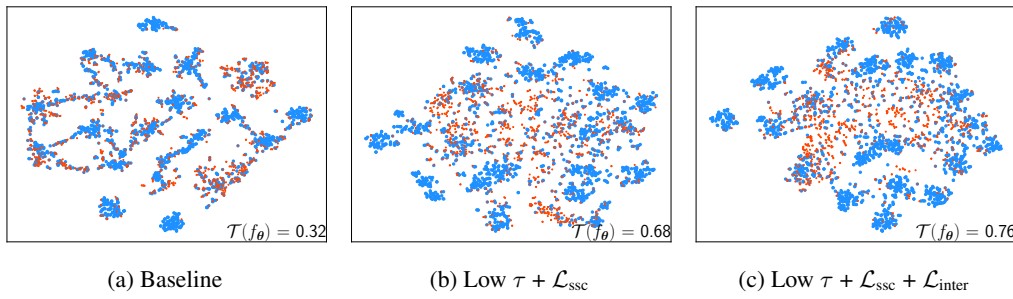

(a) Baseline  (b) Low $\tau + \mathcal{L}_{\text{ssc}}$  (c) Low $\tau + \mathcal{L}_{\text{ssc}} + \mathcal{L}_{\text{inter}}$

Figure 6: T-SNE visualization of learned representations. The blue and red points indicate the base and new class samples, respectively. We measure $\mathcal{T}(f_{\boldsymbol{\theta}})$ to quantify the transferability of the learned representation. We conduct the experiments on CIFAR100 with reduced classes (20 base classes and 10 new classes) for better visualization.

## 4 RELATED WORKS

**Few-Shot Class-Incremental Learning (FSCIL).** Towards the development of real-world artificial intelligence systems, Tao et al. (2020) have initially introduced few-shot class incremental learning. Since its emergence, the field of FSCIL has witnessed significant progresses (Achituve et al., 2021; Cheraghian et al., 2021a;b; Zhu et al., 2021). The early efforts have focused on fine-tuning the feature extractor with the few-shot data of new classes using strong regularization (Tao et al., 2020; Shi et al., 2021; Chen & Lee, 2021) or meta-learning approach (Chi et al., 2022). However, incrementally updating weights with few data inevitably exposes the model to both catastrophic forgetting and overfitting. To address this issue, most of the recent works (Peng et al., 2022a; Liu et al., 2022; Yang et al., 2023; Zhou et al., 2022b; Song et al., 2023; Kalla & Biswas, 2022; Zhuang et al., 2023; Zhang et al., 2021; Hersche et al., 2022) fix the feature extractor trained on base classes and employ a non-parametric classifier using class prototypes (Snell et al., 2017). Since the performance of FSCIL heavily relies on the fixed representation, few works have focused on improving the quality of representation in the context of FSCIL. The common belief on the representation learning for FSCIL have been to encourage greater separation between clusters of base classes in order to reserve the representation space for future new classes. Yang et al. (2023); Hersche et al. (2022) suggest enhancing inter-class separation by enforcing the placement of class prototypes at maximum distances from each other. Zhou et al. (2022a); Song et al. (2023) suggest representation learning with virtual classes in order to promoting both larger inter-class space and a compact intra-class distribution. Conversely, Zou et al. (2022) demonstrate that sharing features between classes can be beneficial to transferability for new classes in the view of margin Wang et al. (2018); Deng et al. (2019).

**Transferable Representation Learning.** The pursuit of learning representations that can be effectively transferred to downstream tasks has gained significant attention in recent years. The early works have focused on supervised training on ImageNet dataset (Russakovsky et al., 2015) and the way to transfer the knowledge to other tasks (Chatfield et al., 2014; Razavian et al., 2014; Donahue et al., 2014). A recent work (Kornblith et al., 2021) demonstrates that the transferability decreases when less features are shared between classes. Subsequently, they suggest lowering the temperature of the softmax cross-entropy loss to encourage the feature sharing to enhance the transferability. Similarly, Liu et al. (2020) suggest using a negative margin in the softmax cross-entropy loss to promote feature sharing rather than focusing on discriminative features among classes. In parallel, several methods have reported the strong transferability of representation learned with self-supervised contrastive learning (Chen et al., 2020; He et al., 2020), in which the features of images and its augmentation counterparts are placed closer while placed far away from those of other images. Islam et al. (2021); Chen et al. (2022) reveal that the spread of features is one possible reason for the enhanced transferability achieved with the self-supervised contrastive loss.

In this work, we investigate the relationship between representation learning and FSCIL that has not yet been explored by previous works on FSCIL or representation learning. In particular, we tackle a widespread belief that a large inter-class distance is beneficial, which is also encouraged by a commonly used loss function, softmax cross-entropy. While there has been few attempts on inducing feature sharing, we show that sharing features within large feature space (a large inter-class distance) is detrimental to FSCIL, leading to our new loss term that explicitly minimizes inter-class distances.

## 5 CONCLUSION

In this work, we focus on representation learning for few-shot class-incremental learning, which is a convoluted problem that requires both transferability and discriminability. Founded on the observation that previous works on representation learning or FSCIL fail to address the trade-off, we present a new loss function to achieve a better transferability-discriminability Pareto front, marching towards better representation learning for FSCIL. To this end, we attribute a trade-off to large distances between classes: trying to share features between classes for transferability results in too much spread of features and degradation of discriminability. Thus, we introduce a inter-class distance minimization loss function to encourage a network to learn compact feature representation, which better facilitates the sharing of features and the learning of fine-grained differences between classes. Strong performance yielded by our representation, the-Closer-the-Better (CnB), underlines the significance of learning representation specialized for FSCIL. Thus, we hope our work will inspire future works in this research direction.

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

Table 5: Comparison between the impact of reducing intra-class and inter-class distance. 'Inter-class*' means that the minimization is carried out on a subset of inter-class pairs, which is chosen to have an equal number as the intra-class pairs, ensuring a fair comparison.

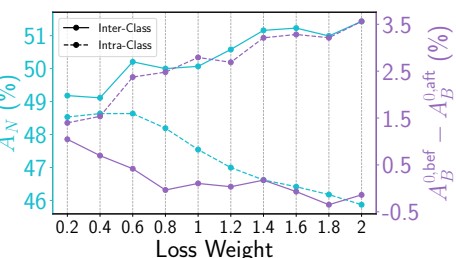

| Dataset | Minimization | $A_B$ | $A_N$ | $A_W$ | $A_B^{0,\text{bef}}$-$A_B^{0,\text{aft}}$ |
|---|---|---|---|---|---|
| CIFAR100 | Intra-class | 68.78 | 24.30 | 50.99 | 2.71 |
| | Inter-class* | 70.32 | 27.28 | 53.10 | **1.31** |
| | Inter-class | **70.72** | **27.23** | **53.32** | 1.58 |
| *mini*ImageNet | Intra-class | 71.17 | 24.10 | 52.34 | 0.43 |
| | Inter-class* | 71.85 | 24.48 | 52.90 | 0.34 |
| | Inter-class | **72.03** | **25.28** | **53.33** | **0.21** |
| CUB200 | Intra-class | 73.12 | 46.45 | 59.63 | 2.58 |
| | Inter-class* | **76.47** | **51.47** | **63.83** | **0.58** |
| | Inter-class | 76.40 | 51.06 | 63.58 | 0.62 |

Figure 7: Comparison between the impact of reducing intra-class and inter-class distance. Reducing intra-class distance harms the performance on new classes (indicated by dotted skyblue line) and exacerbates the performance drop from classifier replacement(indicated by dotted purple line). The experiments are conducted on CUB200 dataset.

## A   IMPLEMENTATION DETAILS

**Self-Supervised Contrastive Learning.** For the self-supervised contrastive learning (SSCL), we generate different views for each image in a mini-batch via data augmentation. For both CIFAR100 and *mini*ImageNet experiments, we apply the random resized cropping, the random horizontal flipping with probability 0.5, and the random AutoAugment Cubuk et al. (2019) with probability 0.5. For CUB200 experiments, we also apply the random resized cropping and the random horizontal flipping with probability 0.5 but without AutoAugment since color information is crucial for fine-grained classification of CUB200 dataset. Unlike the previous methods on SSCL Chen et al. (2020); He et al. (2020), we do not use either a non-linear projection head or a momentum encoder, since we find that the performance difference is not significant.

**Optimization.** For optimization, we use the stochastic gradient descent optimizer with weight decay of $5 \cdot 10^{-4}$ and Nesterov momentum 0.9. We set the initial learning rate as 0.1, 0.1, and 0.005 for CIFAR100, *mini*ImageNet, and CUB200 experiments, respectively, and decay them by 0.1 at the 80% and 90% of the toal training epochs. We set the toal training epochs as 200 for both CIFAR100 and *mini*ImageNet experiments and 50 for CUB200 experiments.

**Hyper-parameters Search Strategy.** In the proposed method, there are 3 hyper parameters including the temperature parameter $\tau$ in the softmax function and the loss weights for the SSCL ($\lambda_{\text{ssc}}$) and the inter-class distance loss ($\lambda_{\text{inter}}$). Since there is no validation datset in the current benchmark setting, we propose a hyper-parameter search strategy using synthetic validation set. The synthetic validation set consists of a set of classes which are obtained by rotating images of base classes similar to Zhang et al. (2021). We observe that the synthesized set have a distinctive distribution compared to the base class distribution, thus it can serve as a validation set.

## B   COMPARISON BETWEEN REDUCING INTRA-CLASS AND INTER-CLASS DISTANCE

As discussed in Section 2.4, the dispersion of representations causes the performance decline in the process of the classifier replacement (CR), since it becomes challenging for the class prototype to adequately represent the scattered cluster. One naïve approach is directly suppressing the intra-class distribution, which can be implemented by minimizing the intra-class distance:

$$\mathcal{L}_{\text{intra}} = -\frac{1}{\sum_{i=1}^{B}\sum_{j>i}^{B}\mathbb{1}_{[y_i=y_j]}}\sum_{i=1}^{B}\sum_{j>i}^{B}\mathbb{1}_{[y_i=y_j]}\text{sim}(\boldsymbol{z}_i, \boldsymbol{z}_j). \qquad (7)$$

However, we empirically find that directly reducing intra-class distance has negative influences on the performance on both base and new classes, as demonstrated in Figure 7 and Table 5. One possible explanation is that reducing intra-class distribution may adversely affect the representation's generalization ability by impeding its capacity to capture fine-grained details among different instances. This

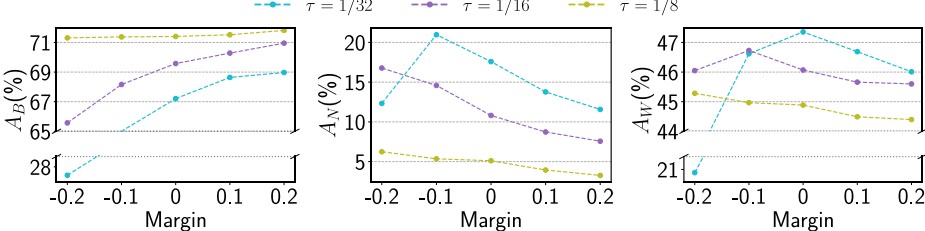

Figure 8: Comparison the impact of the margin and temperature parameters in FSCIL problem. We observe that lowering temperature has a relatively greater influences on the performance than margin. The experiments are conducted on CIFAR100 and we report the averaged results from 3 independent experiments.

is supported by the largely degraded performance results presented in both Figure 7 and the results on CUB200 in Table 5, particularly considering the limited number of training examples in CUB200, which renders a network more prone to overfitting. On the contrary, reducing inter-class distance is observed to achieve outstanding performance both on base and new classes, which validates our assertion on reducing inter-class distance.

## C   COMPARISON BETWEEN MARGIN AND TEMPERATURE

In this section, we compare the effects of the margin parameter ($m$) and the temperature parameter ($\tau$) on representation learning in the context of few-shot class incremental learning (FSCIL). The results in Figure 8 show the accuracy on the whole ($A_W$), base ($A_B$), and new classes ($A_N$) at the end of all training sessions with varying margin and temperature values. As noted in the previous works (Kornblith et al., 2021; Liu et al., 2020; Zou et al., 2022), when the margin and temperature decrease, $A_N$ tends to increase, while $A_B$ tends to decrease (except the case when $m = -0.2$ and $\tau = 1/32$ due to the unstable training). However, we find that the impact of the margin becomes marginal when the temperature is high. For example, when $\tau = 1/8$, the difference between the highest and lowest $A_N$ is roughly 3%, a relatively minor variation compared to the approximately 9% observed with a lower temperature setting. The results with respect to $A_W$ also show that the trade-off between $A_B$ and $A_N$ has a relatively larger correlation with temperature than margin and the highest $A_W$ is achieved when $\tau = 1/32$ and $m = 0$. The feature visualization analysis depicted in Figure 9 also shows similar results that the learned representation is more influenced by the temperature than the margin. In particular, we note that as $\tau$ decreases, it encourages a more dispersed representation, a valuable characteristic for enhancing transferability. Based on this analysis, we regard the temperature parameter as a more effective tool for addressing the issue of base class overfitting and consequently improving transferability.

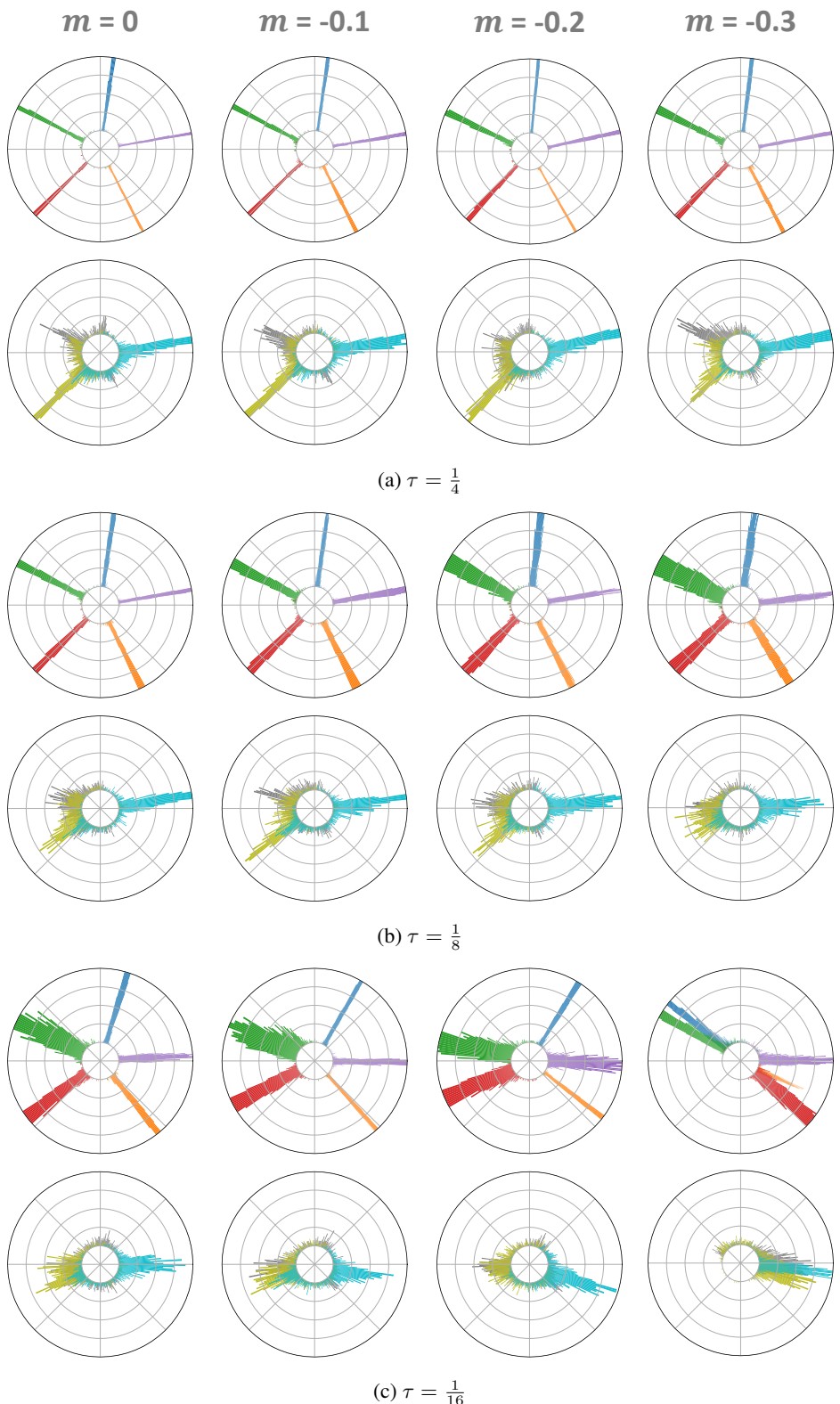

(a) $\tau = \frac{1}{4}$

(b) $\tau = \frac{1}{8}$

(c) $\tau = \frac{1}{16}$

Figure 9: Visualization of representation for comparison the effect of margin and temperature. The results demonstrate that lowering temperature has a relatively greater influences on the performance than margin. We train a network with 2-D feature dimension and visualize angular histograms without dimension reduction. The first and second row in each subfigure indicates the results on base and new classes, respectively. Each color represents a different class.

