# OpenReview forum: "The Closer, The Better: Towards Better Representation Learning for Few-Shot Class-Incremental Learning"
_ICLR.cc/2024/Conference — ICLR 2024 Conference Withdrawn Submission_

### Official Review · Reviewer_fd7A · 2023-10-30

**Soundness:** 3 good
**Presentation:** 2 fair
**Contribution:** 3 good
**Rating:** 5
**Confidence:** 4

**Summary:**

This submission mainly focuses on few-shot class-incremental learning via better representation learning. Traditional approaches observe that representation learning is critical for the transferability and discriminability between all classes. In contrast to current approaches that improve FSCIL via the spread of the intra-class feature representation,  this paper observes that the transferability can harm the discriminability of FSCIL models, as too much spread of features can degrade the quality of the feature-mean class representation in the non-parametric classifier. This paper further introduces to decrease the inter-class distance when increasing the intra-class distance. The experiments demonstrate the effective improvement with the counter-intuitive claim.

**Strengths:**

First of all, the manuscript presents a counter-intuitive but interesting observation for the few-shot class-incremental learning, that is FSCIL requires not only increasing the intra-class distance but also decreasing the inter-class distance.
1. The observation from extensive experiments is interesting and will attract the interest of the community, though I still hold some concerns about this observation.
2. Overall, the analysis is good.

**Weaknesses:**

Though I think the observation is interesting, I have several concerns about this conclusion.

1. The paper is mainly evaluated on small datasets. Therefore, It might be only effective due to the data distribution. For example, if the data has large variations from different domains, it might degrade the performance if it decreases the inter-class distribution.
2. The evaluation is mainly based on small networks, e.g., ResNet-18
3. Considering the weakness of the experiment, the paper also does not have any theory support.

**Questions:**

1. Have you evaluated it on the large datasets? e.g. ImageNet. Or do you try to evaluate the observation when the domain is a bit different from the base classes?
2. Have you tried to evaluate the large-pretrained networks? (e.g. ViT-B (ImageNet), or ViT-L(ImageNet), or CLIP).

---

### Official Review · Reviewer_YAmj · 2023-10-31

**Soundness:** 3 good
**Presentation:** 3 good
**Contribution:** 2 fair
**Rating:** 5
**Confidence:** 4

**Summary:**

The paper introduces a novel approach to representation learning in FSCIL. It observes that improving the transferability of features can harm the discriminability of FSCIL models because too much feature spread can degrade the quality of feature-mean class representations. To address this issue, the paper suggests not only increasing the intra-class distance but also decreasing the inter-class distance. By minimizing the inter-class distance, the learned representation achieves a balance between transferability and discriminability. The proposed representation learning approach, referred to as "the-Closer-the-Better (CnB)," combines the joint objective of softmax cross-entropy (SCE) loss with a lower temperature, self-supervised contrastive (SSC) loss, and inter-class distance minimization. The experimental results demonstrate that CnB improves both transferability and discriminability without requiring the storage of exemplars, test-time augmentation, or weight updates when learning new classes. The paper aims to provide a more effective solution for FSCIL by challenging common beliefs and practices about inter-class distance in representation learning.

**Strengths:**

1. The paper's approach to addressing the challenges of few-shot class-incremental learning (FSCIL) is highly original. It challenges conventional practices by introducing a novel strategy for representation learning that focuses on minimizing the inter-class distance rather than solely maximizing feature spread.
2.The concept of "the-Closer-the-Better (CnB)" represents an innovative perspective on improving the balance between transferability and discriminability in FSCIL. It departs from traditional approaches by offering a counter-intuitive solution to the trade-off between these two aspects.
3.The paper excels in terms of clarity. It is well-written and organized, with a clear introduction and structured sections.

**Weaknesses:**

1.While it claims to have discovered the importance of smaller inter-class distances for maintaining transferability in the context of few-shot class-incremental learning, the paper offers only a simple loss function to address this issue.
2. I think minimizing inter-class distance in a blunt manner may lead to a blurred decision boundary. Additionally, relying solely on hyperparameters to control the L_{inter} and ensure both transferability and discriminability is not practical, especially in long sequences of few-shot incremental learning where old class samples are unavailable, and the number of new samples is limited.
3. The paper highlights the importance of various hyperparameters, such as temperature, loss weights, and margin parameters. It would be beneficial to provide more guidance on how to tune these hyperparameters effectively, as their values can significantly impact the performance of the proposed method.
4. While the paper presents experimental evidence, it lacks a more detailed theoretical analysis of why the proposed approach works. Providing a theoretical foundation for the observed effects, such as the relationship between inter-class distance and transferability, would make the paper more convincing and informative.
5. In addition, there is a problem with the format of the article, and the reference section is incorrectly sized.

**Questions:**

1. Can the authors provide a more in-depth theoretical foundation for the observed effects of minimizing inter-class distance on transferability and discriminability in few-shot class-incremental learning?
2. Given the concerns about the simplicity of the proposed solution, are there plans to explore more complex and diverse methods for addressing the challenges of few-shot class-incremental learning?
3. Given the sensitivity of hyperparameters in the proposed method, can the authors provide guidance on how to effectively tune these hyperparameters for different scenarios and datasets?
4. How does the proposed method impact the interpretability and explainability of learned representations? Can the authors provide insights into which features are selected and shared among classes to enhance model interpretability?

---

### Official Review · Reviewer_h7Ap · 2023-10-31

**Soundness:** 2 fair
**Presentation:** 3 good
**Contribution:** 2 fair
**Rating:** 5
**Confidence:** 2

**Summary:**

In this paper, authors propose to learn transferable and discriminative representations to tackle FSCIL. Under the guidance of empirical results and analysis, authors propose a novel loss function to reduce the inter-class distances. Along with the self-supervised contrastive loss and cross entropy loss, the model further learns features with large intra-class distance.

**Strengths:**

In this paper,  the idea is simple and the proposed method is easy to follow, and the proposed learning objective seems effective on certain datasets.

**Weaknesses:**

Overall, the paper describes the motivations and approaches with experimental results. However, this paper lacks of convincing empirical results of comparing with others to support the proposed method.  Moreover, as the authors mentioned that the minimizing inter-distance is counter-intuitive, I suggest the authors can give more detailed explanation on this, not only show some experimental results to verify the assertion.

More questions,  concerns and suggestions are as follows:
1. Performance on benchmark datasets such as CIFAR100 and MiniImageNet are less convincing for the validity of proposed method. Prevailing methods like NC-FSCIL and SAVC are not included in the CIFAR and MiniImageNet table.

2. The main idea of the paper is learning compact and transferable representations by proposing a novel loss function to minimize the distance between features of all classes.  By utilizing this loss, the model is encouraged to learn similar representations of all classes.  However, is it likely to damage model’s discriminability of novel classes?  In figure 1(c), though proposed method allows the model learns less overlapping representations between base and novel classes, it seems the model cannot separate feature representation of novel classes well compared to baseline methods and self-supervised method in figure (a) and (b). Also in Figure 6(c), the model is not separate the novel classes well.

3. The ablations in table 2 are incomplete. If the model is trained simply uses L_inter or L_ssc without DA or low temperature, which has more impact on the performance?

4. Details of experimental setting are missing. Which dataset is used for experiments in figure 2,3,4? What kind of the incremental learning setting is adopted in these experiments (number of base classes, incremental sessions)?

5. In table 5 second row, Inter-class* obtains the best performance in A_N and should be marked as bold.

6. Under the proposed learning objective, is it likely that CR, which simply computing the mean feature of novel classes, may not be very effective for incremental learning? Without any further constraints on the updated classifier , is it possible that the model learns very similar new class prototypes in incremental sessions, which leads to less discriminative performance on novel classes? Considering other methods that updates the classifier might boost the performance further, such as the mode2, mode3 in Constrained-FSCIL or the NC-FSCIL.

**Questions:**

see the weakness part

---

### Official Review · Reviewer_JXg4 · 2023-11-01

**Soundness:** 3 good
**Presentation:** 3 good
**Contribution:** 2 fair
**Rating:** 3
**Confidence:** 5

**Summary:**

In order to solve the few-shot class-incremental learning (FSCIL) problem, the paper does a fine-grained analysis of transferability, feature spread, and their effects on FSCIL. It, therefore, proposes to decrease the inter-class distance by a weighted inter-class distance minimizing loss, along with the previous self-supervised contrastive loss and CE loss. Experiments on CIFAR100, miniImageNet, and CUB200 demonstrate that the proposed method CnB (the-Closer-the-Better) shows better final accuracy and less performance drop.

**Strengths:**

The paper is well written and easy to follow, with extensive illustration, e.g. feature distribution, T-SNE visualization, etc., and fine-grained analysis.
Ablation experiments and sensitivity analysis of the loss weights help the readers better understand how each component contributes to the final result.

**Weaknesses:**

The novelty of this work seems to be somewhat limited and incremental. The only contribution of the paper is the incorporation of L_inter, an inter-class distance minimizing loss, to decrease the inter-class distance.
The design of additional losses has been extensively studied in related areas such as metric learning, few-shot learning, and continual learning. A unique characteristic of the FSCIL problem is the sequential learning of new classes during the subsequent phases. However, this work only focuses on the pre-training on the base classes and does not provide deeper insight into how the system continuously learns along the time steps.

The improvements by this approach seem to be quite marginal, for instance, 63.58 vs 62.50 on CUB200 (1.08%), and 53.32 vs 52.10 on CIFAR100 (1.22%). It is even significantly worse than ALICE (53.33 vs 55.70) on miniImageNet.
I wonder if the method is tested multiple times with different seeds, can it consistently outperform the other baseline methods, with statistical significance?

**Questions:**

In Tab. 1, Tab. 3, and Tab. 4, the best methods are only denoted as bold for the final phase and the PD. It would be better to also mark these values for the previous phases.

In Fig. 6, the representations in (b) and (c) seem to be less discriminative compared with those in (a). Again, the distributions in (b) (without L_inter) and in (c) do not exhibit significant differences.